METHODS

# SHADE: A multilevel Bayesian framework for modeling directional spatial interactions in tissue microenvironments

Joel Eliason[1]*, Michele Peruzzi[2], Arvind Rao[2,3,4,5]

1 Department of Biomedical Engineering, Johns Hopkins University, Baltimore, Maryland, United States of America, 2 Department of Biostatistics, University of Michigan, Ann Arbor, Michigan, United States of America, 3 Department of Computational Medicine and Bioinformatics, University of Michigan, Ann Arbor, Michigan, United States of America, 4 Department of Biomedical Engineering, University of Michigan, Ann Arbor, Michigan, United States of America, 5 Department of Radiation Oncology, University of Michigan, Ann Arbor, Michigan, United States of America

* jeliaso2@jh.edu

## Abstract

**Motivation:** Understanding how different cell types interact spatially within tissue microenvironments is critical for deciphering immune dynamics, tumor progression, and tissue organization. Many current spatial analysis methods assume symmetric associations or compute image-level summaries separately without sharing information across patients and cohorts, limiting biological interpretability and statistical power.

**Results:** We present SHADE (Spatial Hierarchical Asymmetry via Directional Estimation), a multilevel Bayesian framework for modeling asymmetric spatial interactions across scales. SHADE quantifies direction-specific cell-cell associations using smooth spatial interaction curves (SICs) and integrates data across tissue sections, patients, and cohorts. Through simulation studies, SHADE demonstrates improved accuracy, robustness, and interpretability over existing methods. Application to colorectal cancer multiplexed imaging data demonstrates SHADE's ability to quantify directional spatial patterns while controlling for tissue architecture confounders and capturing substantial patient-level heterogeneity. The framework successfully identifies biologically interpretable spatial organization patterns, revealing that local microenvironmental structure varies considerably across patients within molecular subtypes.

## Author summary

The spatial arrangement of cells within tumors provides critical insights into cancer progression and treatment response. Modern imaging technologies can map cellular neighborhoods across multiple tissue sections from many patients, but almost all existing statistical methods face at least one of two key limitations: they

**Data availability statement:** Open-source implementation of this method is available at https://github.com/jeliason/SHADE and code to reproduce our analyses is available at https://github.com/jeliason/shade_paper_code. The data used in this study are publicly available from the Mendeley Data repository at https://data.mendeley.com/datasets/mpjzbtfgfr/1.

**Funding:** JE was funded by NIH-NCI 5 R01-CA268426–03, R37CA214955-01A1, NSF Award Number 215776 and Advanced Proteogenomics of Cancer (T32 CA140044). AR was supported by NIH-NCI 5 P30 CA046592, a gift from Agilent Technologies, and a Precision Health Investigator award from UM Precision Health, NCI Grant R37-CA214955, the University of Michigan (UM) startup institutional research funds and a Research Scholar Grant from the American Cancer Society (RSG-16-005-01). The funders had no role in study design, data collection and analysis, decision to publish, or preparation of the manuscript.

**Competing interests:** We have read the journal's policy and the authors of this manuscript have the following competing interests: AR serves as a member for Voxel Analytics LLC and consults for Telperian, Tempus Inc. and TCS Ltd.

assume spatial relationships are symmetric (if immune cells cluster near tumor cells, tumor cells must cluster near immune cells), and/or they analyze each tissue section independently rather than pooling information across the biological hierarchy. We developed SHADE (Spatial Hierarchical Asymmetry via Directional Estimation) to address both challenges simultaneously. SHADE captures directional spatial dependencies, allowing asymmetric relationships between cell types, while operating within a multilevel Bayesian framework that borrows strength across tissue sections, patients, and patient groups. This hierarchical structure yields more precise estimates and directly quantifies variability at each biological scale. Applying SHADE to colorectal cancer data revealed distinct directional spatial patterns distinguishing immune-infiltrated from immune-excluded tumors, with substantial patient-level heterogeneity indicating diverse microenvironment architectures within disease subtypes. SHADE provides a principled approach for analyzing the directional, multi-scale organization of tissue microenvironments.

## 1 Introduction

Spatial dependencies between cell types play a central role in immune dynamics, tumor behavior, and tissue organization, motivating statistical models that can capture such interactions [17,29]. The tumor microenvironment (TME) is a spatially structured system where cell arrangements are closely linked to disease progression and treatment response [8,21]. Notably, these spatial interactions are frequently asymmetric: immune cells may cluster near tumor cells without the reverse being true, reflecting directional dependencies in tissue organization [7].

Recent advances in multiplexed imaging technologies, including multiplexed immunofluorescence (mIF) and other high-resolution spatial profiling methods, have made it possible to quantify cell type spatial distributions and interactions within the tumor microenvironment at single-cell resolution [23]. Standard pipelines for analyzing spatial cellular interactions in multiplexed imaging data typically compute second-order summary statistics, such as Ripley's $K$- and $L$-functions or the $G$-cross function, independently for each image to quantify tumor–immune interactions, infiltration patterns, or neighborhood structure [5,9,14,15,20,22,24,27]. These per-image summaries are then compared across groups or used as covariates in downstream analyses. Although effective for descriptive comparisons, this approach treats each image as an isolated point pattern and cannot share information across biological replicates during estimation. Multilevel models for replicated point patterns have been proposed [4,6,13,16,19,28], but they typically model derived summaries via two-stage estimation or use parametric models fit via pseudolikelihood with random effects, rather than jointly estimating flexible interaction functions through full Bayesian multilevel inference at the point-process level.

Beyond these methodological limitations in handling replication, existing spatial models also struggle with directional dependencies. Gibbs point process models provide a flexible probabilistic framework [3,18], yet standard formulations assume symmetric interactions. While so-called hierarchical Gibbs models have been proposed

to accommodate directionality [11,12], standard implementations depend on parametric interaction functions that impose restrictive assumptions. Observations based on the *G*-cross function suggest that spatial interactions between cell types are inherently asymmetric [26], motivating statistical models that directly account for directional spatial effects.

In this work, we develop SHADE (Spatial Hierarchical Asymmetry via Directional Estimation), a statistical framework for modeling asymmetric spatial associations in tissue microenvironments. Our primary contribution is methodological: extending multitype Gibbs point process models [3,12,18] by modeling directional associations via *spatial interaction curves (SICs)* that quantify how the presence of one cell type affects the expected density of another using flexible basis expansions within a multilevel Bayesian structure. The hierarchical framework enables partial pooling across biological scales (images, patients, cohorts) with full posterior inference for uncertainty quantification and heterogeneity analysis. For computational efficiency, we use logistic regression with quadrature-based dummy points rather than direct Poisson likelihood estimation [2]. We validate SHADE's performance through comprehensive simulation studies and demonstrate its utility on colorectal cancer data, where it successfully quantifies spatial organization patterns while controlling for tissue architecture confounders. Technical comparison with Gibbs models is provided in Sect A in S1 Text. An overview of the SHADE workflow is provided in Fig 1.

## 2 Methods

### 2.1 Multilevel modeling of conditional spatial point processes

We model the spatial distribution of a target cell type *B* given the presence of one or more conditioning cell types $A_1, A_2, \ldots, A_K$, using a hierarchical framework based on conditional spatial point processes. Our approach extends hierarchical Gibbs models [11,12] and is designed to flexibly estimate asymmetric spatial association patterns while accounting for multilevel variation across images, patients, and cohorts.

Let $X_{A_k}, X_B \subset W$ denote the observed spatial point patterns of cell types $A_k$ and $B$, respectively, within a two-dimensional tissue region $W \subset \mathbb{R}^2$. Formally, $X_{A_k} = \{x_i^{(k)} \in W \mid i = 1, \ldots, N_{A_k}\}, \quad X_B = \{y_j \in W \mid j = 1, \ldots, N_B\}$, where $x_i^{(k)} = (x_{i1}^{(k)}, x_{i2}^{(k)}) \in \mathbb{R}^2$ denotes the two-dimensional spatial coordinate of the *i*-th cell of type $A_k$, and $y_j = (y_{j1}, y_{j2}) \in \mathbb{R}^2$ denotes the coordinate of the *j*-th cell of type *B*. The quantities $N_{A_k}$ and $N_B$ indicate the total number of observed cells of type $A_k$ and $B$, respectively. The observation window *W* corresponds to the region of tissue captured in the image, typically a rectangular subset of the plane defined by the image dimensions.

In practice, $X_{A_k}$ and $X_B$ are obtained from image segmentation and cell type classification pipelines applied to high-resolution tissue images, such as those generated by multiplexed imaging platforms.

We model the spatial distribution of $X_B$ (the *target* cell type, e.g., immune cells) conditional on $X_{A_1}, \ldots, X_{A_K}$ (the *source* cell types, e.g., tumor cells and vasculature) by assuming that $X_B \mid X_{A_1}, \ldots, X_{A_K}$ follows an inhomogeneous Poisson point process [3]. This allows the expected density of target cells to vary flexibly across space as a function of source cell locations and covariates, enabling the framework to represent complex spatial patterns including clustering, repulsion, and distance-dependent associations. While alternative models exist (e.g., Cox processes and cluster processes), the inhomogeneous Poisson process provides the best balance of flexibility, interpretability, and computational feasibility for our application [3]. The likelihood is:

$$
L(X_B \mid X_{A_1}, \ldots, X_{A_K}) = \left[ \prod_{v \in X_B} \lambda(v \mid X_{A_1}, \ldots, X_{A_K}) \right]
$$
$$
\times \exp\left( -\int_W \lambda(v \mid X_{A_1}, \ldots, X_{A_K}) \, dv \right),
$$

(1)

where the product runs over all observed locations of type *B*, while the integral accounts for the total expected intensity over the observation window *W*. The point process for $X_B$ is entirely characterized by its *conditional intensity function*

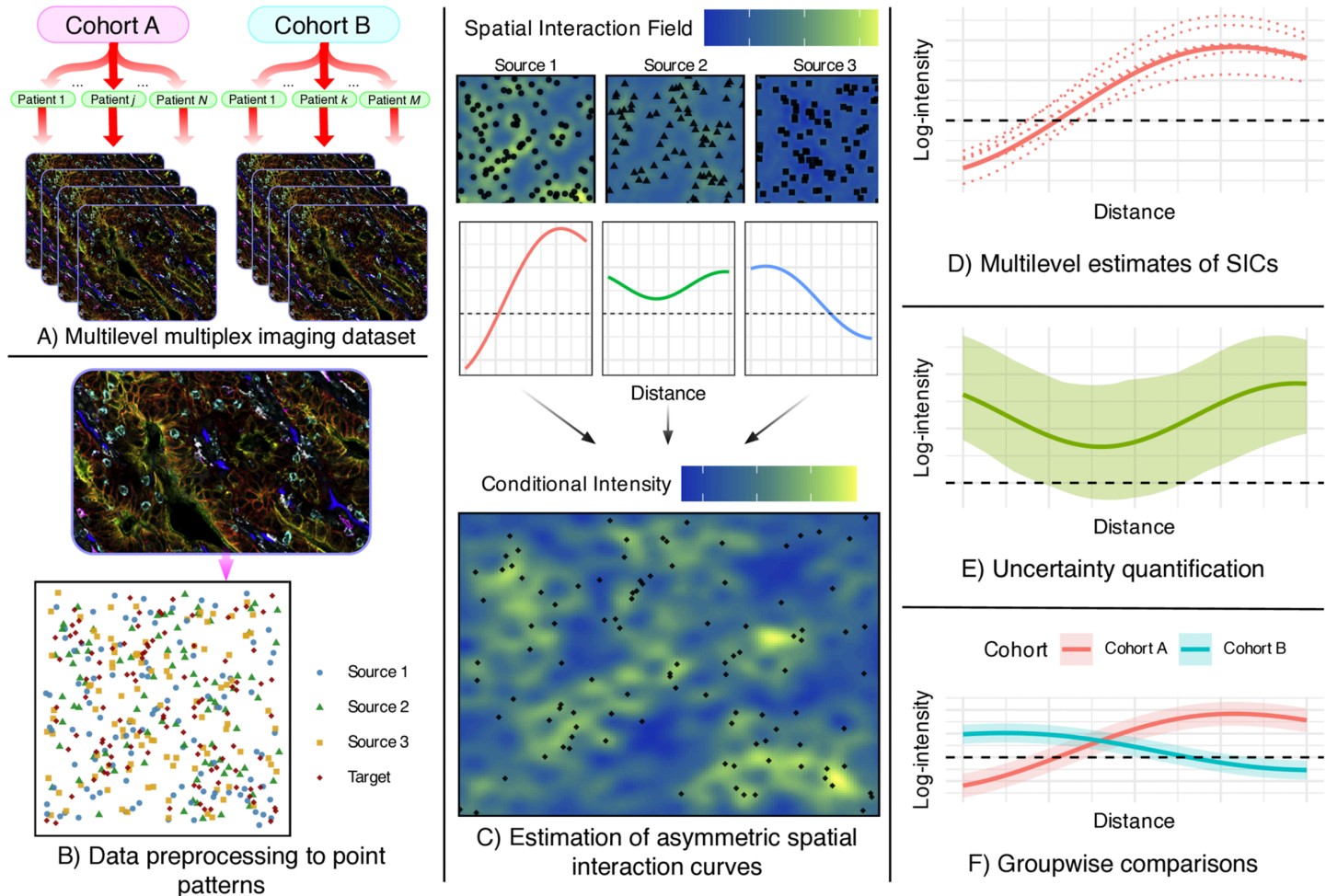

**Fig 1.** **Summary of the SHADE (Spatial Hierarchical Asymmetry via Directional Estimation) framework. A)** Multiplexed imaging data is structured hierarchically across cohorts, patients, and images. Multiplex images were adapted from [21] and are used under a Creative Commons CC BY 4.0 license. **B)** Images are processed into spatial point patterns with cell type annotations. **C)** SHADE estimates Spatial Interaction Curves (SICs) that capture directional associations between cell types across spatial scales. **D)** SICs are estimated at cohort, patient, and image levels, enabling multilevel analysis of spatial heterogeneity. **E)** Posterior distributions provide uncertainty quantification. **F)** SICs can be compared across cohorts to assess differences in spatial organization.

$\lambda(v \mid X_{A_1}, \ldots, X_{A_K})$, which defines the expected local density of type $B$ cells at any location $v \in W$, conditioned on the spatial configurations of the conditioning cell types, that is, $\lambda(v \mid X_{A_1}, \ldots, X_{A_K}) = \lim_{|dv| \to 0} \frac{1}{|dv|} E[N_B(dv) \mid X_{A_1}, \ldots, X_{A_K}]$.

We model $\lambda(\cdot)$ as depending log-linearly on the observed patterns of $A_1, \ldots, A_K$ cells:

$$\log \lambda(v \mid X_{A_1}, \ldots, X_{A_K}) = \beta_0 + \mathbf{z}^\top(v)\beta + \sum_{k=1}^{K} \mathbf{q}_{A_k}^\top(v)\psi_{A_k} \qquad (2)$$

In (2), the term $\mathbf{z}(v) \in \mathbb{R}^J$ is a vector of covariates at location $v$, with corresponding coefficients $\beta \in \mathbb{R}^J$. The vector $\mathbf{q}_{A_k}(v) \in \mathbb{R}^P$ encodes spatial interaction features between type $A_k$ and type $B$; its $p$-th element is defined as:

$$[\mathbf{q}_{A_k}(v)]_p = \sum_{x \in X_{A_k}} \phi_p(\text{dist}(v, x)), \qquad (3)$$

where $\phi_p(\cdot)$ is a basis function (e.g., a B-spline or Gaussian kernel) that modulates the influence of a type $A_k$ cell at a given distance from $v$ and $\psi_{A_k} \in \mathbb{R}^P$ are the corresponding coefficients. Finally, the intercept term ($\beta_0$) captures baseline log-intensity, effectively normalizing to average target cell density. SIC values then quantify deviations from this baseline as a function of proximity to source cells, ensuring curves are directly comparable across images and patients despite differences in overall cell abundance. This captures how proximity to different cell types influences target cell density across spatial scales. The smooth basis functions allow flexible, distance-dependent associations, generalizing traditional Gibbs models that use fixed interaction radii [1,11].

The basis coefficients $\psi_{A_k}$ from Eq (2) are not directly interpretable. To understand how spatial associations change with distance, we define the *spatial interaction curve* (SIC), which summarizes how proximity to source cell type $A_k$ affects the expected density of target cell type $B$ at each distance.

**2.1.1 Spatial interaction curve.** The SIC summarizes the asymmetric spatial association between a conditioning cell type $A_k$ and a target cell type $B$ as a function of distance $s$:

$$\text{SIC}_{A_k \to B}(s) = \sum_{p=1}^{P} \psi_{A_k}^{(p)} \phi_p(s), \tag{4}$$

This curve represents the expected contribution of type $A_k$ cells to the log-intensity of type $B$ cells as a function of distance $s$ from a type $A_k$ cell. Here, log-intensity refers to the logarithm of the conditional intensity function $\lambda(v)$ (Eq 2), which describes the expected spatial density of type $B$ cells (in cells per unit area) at any location $v$. Because the model is log-linear, SIC values quantify additive changes on the log-intensity scale, which correspond to multiplicative changes in actual cell density. For example, SIC = 0.2 at distance $s$ implies an $e^{0.2} \approx 1.22\times$ (or 22%) increase in the expected local density of type $B$ cells at radius $s$ from a type $A_k$ cell, while SIC = $-0.3$ corresponds to an $e^{-0.3} \approx 0.74\times$ decrease (26% reduction). Fig 2 illustrates how SIC values correspond to spatial attraction (positive) or repulsion (negative).

A key advantage of our framework is that the spatial interaction terms $\psi_{A_k}$ can take both positive and negative values, allowing for flexible modeling of attraction and repulsion. This mirrors the flexibility of hierarchical Gibbs models

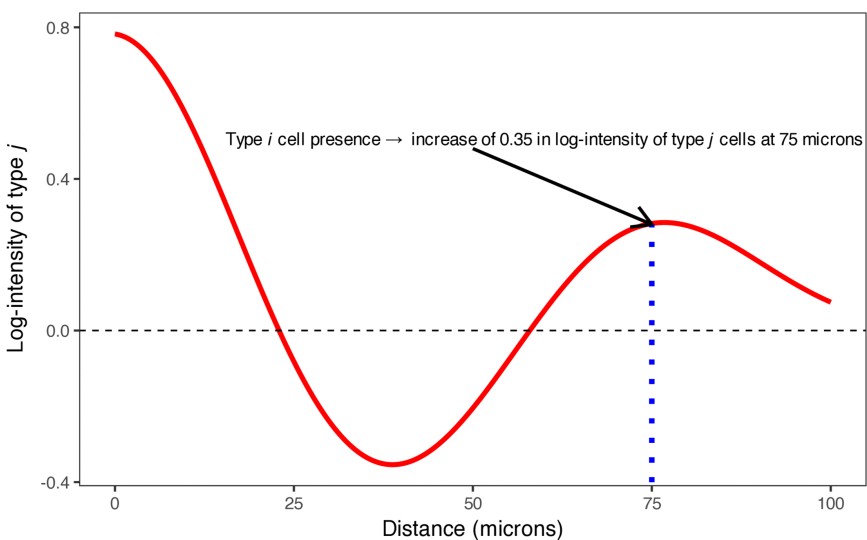

**Fig 2. An example spatial interaction curve showing the effect of a source cell type $A_k$ on a target cell type B.** At each distance $s$, the curve value represents the change in log-intensity of type $B$ associated with the presence of a type $A_k$ cell at distance $s$.

while overcoming constraints in symmetric pairwise interaction models, which typically restrict interaction terms to be non-positive and therefore cannot directly model spatial attraction.

Because cell centroids cannot occur arbitrarily close in space, very short distances (below approximately 1-2 cell diameters) can reflect geometric crowding, cell-cell contact, or segmentation artifacts, making biological interpretation ambiguous. We therefore predefine a minimum interaction radius $r_{min} = 25\ \mu$m (approximately 1.5-2 typical cell diameters) and report spatial interaction curves only for $r \geq r_{min}$. The model is fit using all observed cell locations, but posterior summaries and spatial interaction curves are evaluated and reported only at distances $r \geq r_{min}$ to focus interpretation on unambiguous intercellular spacing. All band-level posterior probabilities are computed on intervals $I \subseteq [r_{min}, \infty)$.

**2.1.2 Interpreting spatial interaction curves.** Our modeling framework captures *predictive* spatial associations, not biological causation. A strong $A \rightarrow B$ SIC indicates that type $A$ cell locations are statistically predictive of local type $B$ density, formally meaning that the conditional intensity $\lambda(v \mid X_A)$ differs from the marginal intensity $\lambda_B(v)$. While SICs quantify the strength and direction of spatial predictability, they do not establish causal biological effects. The model can incorporate spatial covariates (e.g., distance to tumor margin) to adjust for confounding from first-order intensity effects.

## 2.2 Multilevel Bayesian model

To capture biological variability across individuals and sampling levels, we impose a hierarchical prior structure on the spatial interaction coefficients. For each conditioning cell type $A_k$, the interaction coefficients are indexed hierarchically across three levels: cohort ($\psi$), patient ($\gamma$), and image ($\delta$). Specifically, for each basis function $p$, we define:

$$
\begin{aligned}
\psi_{A_k}^{(g,p)} &\sim \mathcal{N}(0, \sigma_{cohort}^2), \\
\gamma_{A_k}^{(n,p)} &\sim \mathcal{N}(\psi_{A_k}^{(g(n),p)}, \sigma_{patient}^2), \\
\delta_{A_k}^{(m,p)} &\sim \mathcal{N}(\gamma_{A_k}^{(n(m),p)}, \sigma_{image}^2),
\end{aligned}
\tag{5}
$$

where $g(n)$ maps patient $n$ to its cohort, and $n(m)$ maps image $m$ to its corresponding patient. Each image-level coefficient $\delta_{A_k}^{(m,p)}$ governs the localized effect of source cell type $A_k$ on target cell type $B$ at distance scale $p$, within image $m$.

This hierarchical structure allows partial pooling across images, patients, and cohorts, improving estimation accuracy while capturing real biological variation in spatial interactions. The model can estimate both shared patterns across groups and context-specific effects within individual patients or images. When comparing spatial organization across biological groups (e.g., treatment responders vs. non-responders, different tumor subtypes), each group is modeled as a separate cohort with its own cohort-level parameters $\psi_{A_k}^{(g,p)}$. Differences between groups are then assessed by comparing the posterior distributions of these cohort-level parameters, as quantified through SICs (Eq 4) and their associated simultaneous credible bands.

Hyperpriors for the variance components $\sigma_{cohort}^2$, $\sigma_{patient}^2$, and $\sigma_{image}^2$ are detailed in Sect F in S1 Text.

## 2.3 Uncertainty quantification and prioritization of cell type pairs

To quantify uncertainty in estimated SICs and assess statistical significance, we employ simultaneous 95% credible bands that account for multiple comparisons across the distance domain. Unlike pointwise credible intervals, simultaneous bands provide joint coverage across all distances within a specified range, offering stronger protection against false discoveries. Statistical significance can be assessed by examining whether the simultaneous band excludes zero over a distance range of interest. Full implementation details are provided in Sect D.1 in S1 Text.

In exploratory analyses involving many cell type pairs ($K(K-1)$ directed pairs for $K$ cell types), we propose summary measures to facilitate prioritization: peak location and magnitude (identifying where the strongest interaction occurs), persistence over biologically relevant distance ranges (quantifying consistent associations within pre-specified intervals), and overall strength (integrating absolute effect sizes over significant regions). These measures can be computed across all

source–target pairs and visualized as heatmaps for systematic comparison. Formal definitions are provided in Sect D.2 in S1 Text.

## 2.4 Logistic regression approximation for computational efficiency

Direct estimation of the Poisson likelihood in spatial point process models requires numerical integration over fine spatial grids, which becomes computationally expensive and unstable. Following [2], we use a logistic regression approximation that introduces dummy points sampled from a homogeneous Poisson process, reframing spatial intensity estimation as binary classification. This approach avoids computational challenges such as singular design matrices and biased uncertainty estimates that arise in direct Poisson modeling. This approximation does not require the target cell process to be homogeneous—only the dummy points are placed homogeneously for computational convenience. Target cells are still modeled as an inhomogeneous Poisson process with spatial variation captured through distance-based SIC features $q_{A_k}(v)$ and spatial covariates $z(v)$ (Eq 2). In our colorectal cancer analysis, we demonstrate this flexibility by incorporating compartment indicators to control for tissue architecture (Comparison of spatial interactions between immune-infiltrated and immune-excluded tumors), showing how covariates can disentangle spatial interactions from architectural confounding. Details of the approximation are provided in Sect B in S1 Text.

## 2.5 Model estimation and computational implementation

Model fitting is implemented in `Stan` using Hamiltonian Monte Carlo (HMC) via `cmdstanr` [10,25].

To approximate the likelihood, we first generate a set of dummy points $D$ from a homogeneous Poisson process with intensity $\lambda_{dummy}$ over the observation window $W$. For each location $v \in X_B \cup D$, we then compute spatial covariates $z(v)$ and interaction features $q_{A_k}(v)$, where each interaction feature encodes basis-function-weighted distances to cells of type $A_k$.

Feature construction for each target cell type $B$ involves evaluating inter-cell distances between observed and dummy target locations and all non-target cells. This step scales as $\mathcal{O}(n_{target} \times n_{source})$, which is quadratic in the total number of cells when the target and source sets are of similar size. However, this operation is implemented using the optimized `crossdist` routine from `spatstat.geom`, which efficiently computes pairwise distances in compiled code. The resulting distance matrix is reused across all basis functions $\{\phi_p\}$ and source types $\{A_k\}$, so the dominant cost occurs only once per target type.

To assess the practical runtime implications, we performed a timing experiment varying the total number of simulated cells from 5,000 to 250,000 and fitting the model via variational inference (Sect D.8 in S1 Text). Feature construction time scaled as $O(n^{1.46})$ (empirical exponent from log-log regression), while total model fitting time scaled as $O(n^{0.85})$ due to efficient distance matrix reuse. At 100,000 cells, total fitting time was approximately 36 seconds; at 250,000 cells, approximately 133 seconds. These benchmarks demonstrate that SHADE remains computationally tractable for large-scale multiplexed imaging studies.

Using these constructed features, we fit a multilevel Bayesian logistic regression model based on the approximation in Eq (10) in S1 Text, with spatial interaction coefficients $\delta_{A_k}^{(m,p)}$ modeled hierarchically according to the priors in (5). Finally, we extract posterior draws of the interaction coefficients and reconstruct the spatial interaction curves using (4).

## 3 Simulation studies

To evaluate SHADE's performance, we conducted simulation studies generating synthetic spatial point patterns with asymmetric interactions and multilevel structure. Spatial patterns were simulated within a bounded domain $W = [0, S] \times [0, S]$, with source points from a homogeneous Poisson process and target points influenced by spatial interaction curves defined over source-target distances. Hierarchical structure was introduced via interaction coefficients generated according to (5).

The final point pattern was converted into a logistic regression dataset using dummy points sampled from a homogeneous Poisson process with intensity $\lambda_{dummy}$, and spatial interaction features were computed as in (3). We first validated that SHADE's hierarchical structure substantially improves estimation quality compared to non-hierarchical alternatives (Sect D.4 in S1 Text). We then conducted a comprehensive comparison of SHADE's detection capabilities against established spatial analysis methods across varying data conditions, presented below. Additional hyperparameter studies are provided in Sect D.5 in S1 Text.

### 3.1 Comparison of spatial pattern detection accuracy across methods

We evaluated SHADE's detection power and calibration by generating hierarchical point patterns with known positive interactions between two cell types.

These simulations evaluate image-level detection, that is, identifying spatial interactions within individual tissue sections. We compare SHADE against *G*-cross and *K*-cross envelope tests based on Monte Carlo simulations under complete spatial randomness. Sect E.7.3 in S1 Text compares SHADE's group-level inference with functional data analysis methods.

We simulated hierarchical point patterns (40 patients, 1–3 images per patient) with positive spatial interactions at multiple distance scales. We varied source cell density (15 vs. 150 cells per image) and target cell density (15 vs. 150 cells per image) across 50 replicates per condition. This assesses how SHADE's hierarchical pooling performs when conditioning information is sparse versus abundant. Detection power is the proportion of simulations where the method correctly identified non-zero spatial interactions in the 0–75 $\mu$m range. For SHADE, detection was based on whether 95% simultaneous credible bands (Sect D.1 in S1 Text) excluded zero at any distance. For comparison, we evaluated three baseline approaches: (1) *G*-cross envelope tests [3], testing whether the observed nearest-neighbor distribution falls outside 95% global envelopes constructed from 99 completely spatially random (CSR) simulations; (2) *K*-cross (specifically, *L*-cross) envelope tests, using cumulative counts rather than nearest-neighbor distances; and (3) a 'Flat' model that estimates SICs independently for each image without hierarchical pooling. Type I error rates were assessed using null simulations with zero spatial interactions. Fig 3 illustrates one simulated example; full simulation details are provided in Sect D.5 in S1 Text.

Fig 4 shows detection power across simulation conditions. SHADE Hierarchical's performance depends critically on two factors: source cell density (which provides conditioning information) and the number of images available for hierarchical pooling.

First, when source density is high (regardless of target density), SHADE achieves excellent median power (100%) across all conditions, substantially outperforming envelope tests when target density is low (SHADE 100% vs. G-cross 77%, K-cross 73%). Abundant source cells provide strong conditioning information that hierarchical pooling can effectively leverage. Conversely, when source density is low, performance depends on whether multiple images are available: with 2–3 images per patient, median power reaches 100%; with only 1 image, power drops to 31% as limited conditioning information per image prevents effective pooling.

SHADE Hierarchical requires at least 2 images per patient for stable performance. With only 1 image per patient, the method exhibits extreme behavior: overly conservative under some conditions (100% coverage, 0% type I error when target density is high and source density is low) and unreliable under others (0% power with high type I error variability when both densities are low). With 2–3 images, performance stabilizes substantially, achieving high power when source density is adequate while maintaining reasonable calibration.

When both source and target densities are low, all methods struggle. With 2–3 images per patient, SHADE achieves 26–28% median power with 94% coverage and well-controlled type I error (4%), demonstrating appropriate conservatism (SHADE Flat has higher power but much worse coverage in this regime, in the multi-image case - see Fig K in S1 Text). Interestingly, *G*-cross performs best in this regime (49% power).

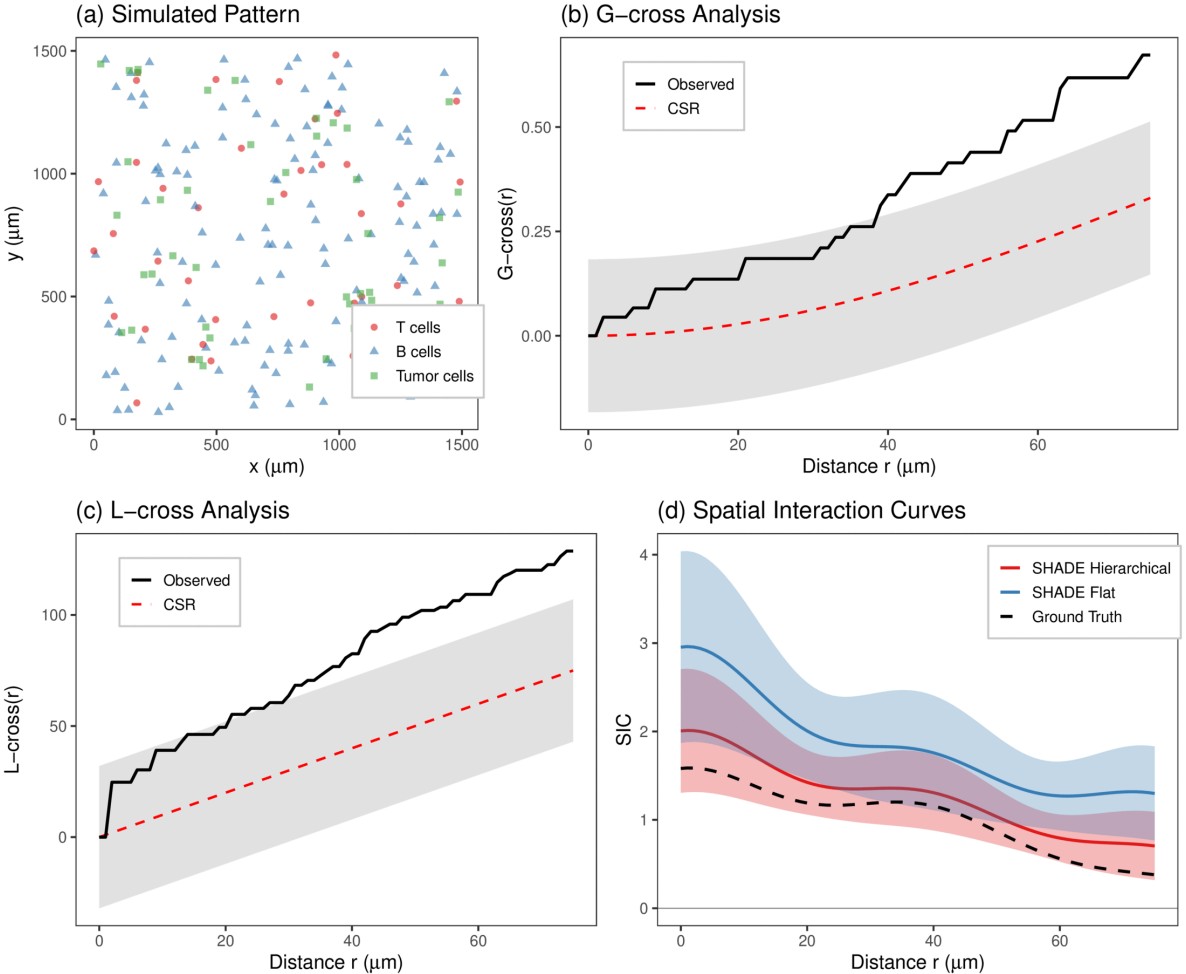

**Fig 3**. **Simulation study comparing spatial analysis methods.** (a) Simulated pattern showing T cells, B cells, and tumor cells in a 1500 × 1500 $\mu m^2$ region with known spatial clustering. (b) *G*-cross analysis with the observed curve (black), CSR expectation (dashed red), and 95% global envelope (gray ribbon) from 99 CSR simulations. (c) *L*-cross analysis, which counts all tumor cells within distance *r* of a typical T cell rather than measuring nearest-neighbor distances. (d) SIC estimates from SHADE Hierarchical (red), SHADE Flat (blue), and the ground truth (black dashed), with simultaneous 95% credible bands.

Coverage and type I error performance (Sect D.9 in S1 Text) reveal adaptive calibration. When source density is high (favorable for detection), SHADE trades calibration for sensitivity (55% coverage, 100% power, <1% type I error). When both densities are low (unfavorable for detection), SHADE becomes appropriately conservative (94% coverage, 28% power, 4% type I error). This contrasts with SHADE Flat, which maintains poor coverage (73%) regardless of scenario and shows severely inflated type I error rates (28%). Median type I error rates for SHADE Hierarchical with 2–3 images are well-controlled (0.8–7.5%), comparable to envelope tests (G-cross: 2.5–8.8%; K-cross: 1.3–6.7%), though with higher variability (IQR up to 0.12), indicating occasional liberal inference under challenging conditions.

**3.1.1 Robustness to spatial confounding.** We also tested SHADE's performance when the model is misspecified due to unmeasured spatial heterogeneity—specifically, discrete tissue compartments (e.g., tumor islands, stromal regions) that create baseline density differences independent of source-target interactions (Fig 5; Sect D.10 in S1 Text). Results reveal regime-dependent bias: when both cell types are abundant, SHADE achieves perfect detection power but exhibits

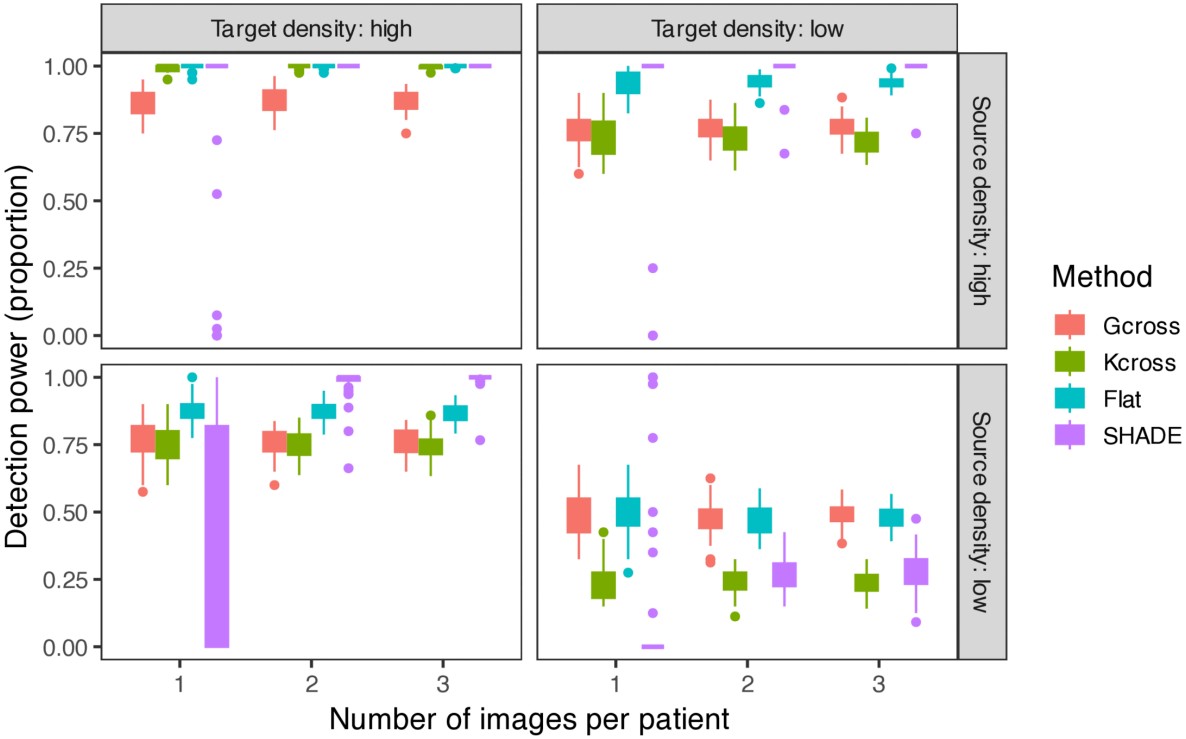

**Fig 4**. **Detection power comparison across simulation conditions.** Boxplots show the proportion of all images in which methods correctly identify non-zero spatial interactions by testing whether simultaneous credible bands (SHADE) or global envelopes (G-cross, K-cross) exclude zero anywhere in the 0–75 $\mu$m range. Results are stratified by source cell density (rows: the conditioning cell type) and target cell density (columns: the cell type being modeled), with number of images per patient (1, 2, or 3) shown on the x-axis. SHADE Hierarchical achieves highest power when source density is high, with performance when source density is low depending critically on having multiple images per patient available for hierarchical pooling (see main text).

elevated Type I error rates (11.7–17.1%) and severely undercovers (43–52% vs. expected 95%), incorrectly attributing compartment effects to source-target interactions. When target density is low, wider credible bands provide partial robustness (82–93% coverage, 1.7–5.8% Type I error). These findings indicate that unmeasured spatial structure can produce substantial bias in high-density scenarios, suggesting the need for explicit compartment modeling or sensitivity analyses when such heterogeneity is suspected.

## 4 Results: Multiscale inference of directional spatial interactions in colorectal cancer

### 4.1 Description of colorectal cancer dataset

We applied SHADE to a colorectal cancer dataset [21] of 35 patients (140 images) stratified by immune phenotype: Crohn's-like reaction (CLR, immune-infiltrated) and diffuse inflammatory infiltration (DII, immune-excluded). We analyzed eight cell types with three target populations (CD8[+] T cells, memory CD4[+] T cells, granulocytes) and five source populations (vasculature, tumor cells, cancer-associated fibroblasts [CAFs], tumor-associated macrophages [TAMs], hybrid epithelial-mesenchymal [E/M] cells). Details on cell type reclassification and data preparation are in Sect E.1 in S1 Text.

### 4.2 SHADE characterizes multiscale spatial organization patterns

We extracted image-, patient-, and cohort-level interaction parameters and computed SICs as in Eq 4. SHADE identified three key spatial phenomena. First, negative associations at <25 $\mu$m across nearly all pairs reflect physical

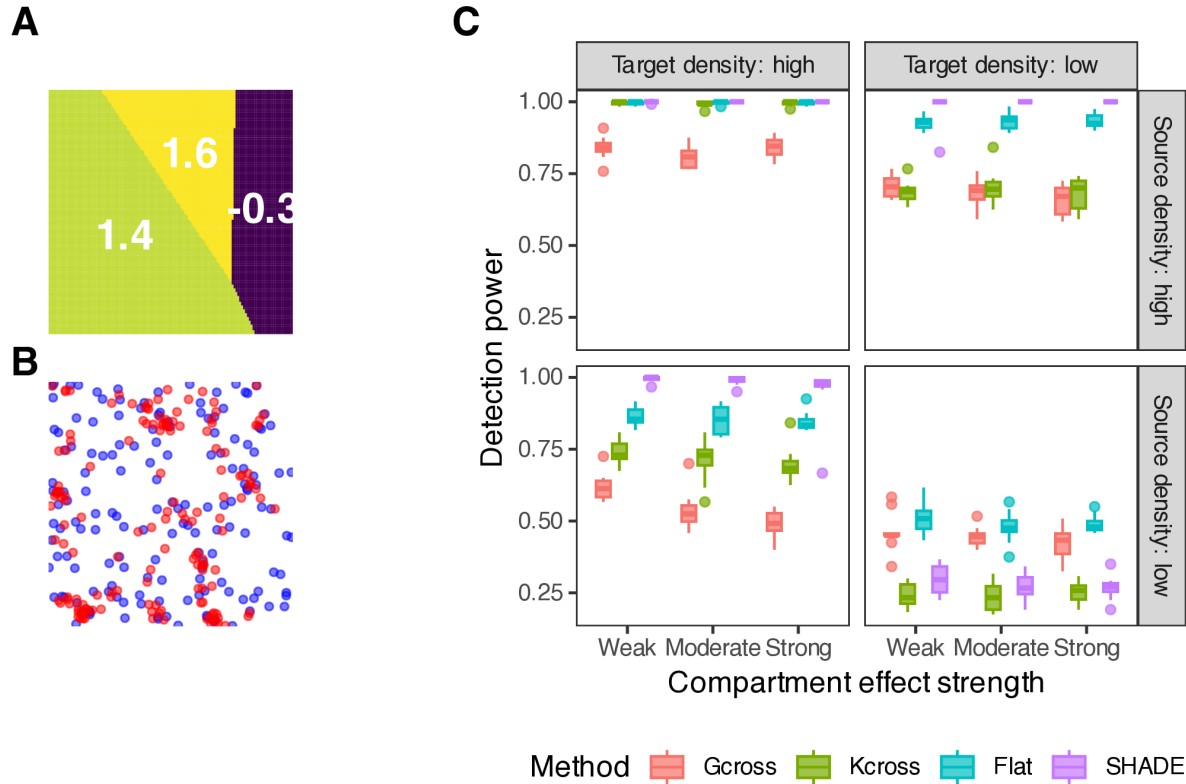

**Fig 5. Robustness to spatial confounding via compartments.** A: Compartment structure showing log-intensity effect on target density (3 compartments with moderate effect strength). B: Example simulated pattern with tumor cells (red) and T cells (blue). C: Detection power stratified by compartment effect strength (weak/moderate/strong), source density (T cells, rows), and target density (tumor cells, columns). Despite unmeasured compartments, all methods maintain high power in favorable scenarios. However, elevated Type I error rates (see Sect D.10 in S1 Text) indicate that SHADE incorrectly attributes compartment effects to source-target interactions when both cell types are abundant, demonstrating regime-dependent confounding bias.

crowding. Second, many pairs showed peaks or valleys at 25–75 $\mu$m consistent with active coordination. For example, Fig 6 shows CTLs in CLR patients avoiding CAFs (log-intensity < 0), while DII patients show neutral to clustering patterns. Third, patient-specific SICs (dashed lines, Fig 6) reveal substantial within-group variation beyond molecular subtype classifications. Associations at >75 $\mu$m were generally weaker or absent.

### 4.3 Quantifying spatial heterogeneity across biological scales

Unlike descriptive spatial statistics that compute separate summaries for each image, SHADE's hierarchical Bayesian model jointly estimates SICs at the image, patient, and cohort levels through partial pooling. This enables formal decomposition and quantification of variability at each biological scale, allowing us to assess both intra- and inter-patient variability and distinguish conserved from patient-specific spatial patterns.We quantified between-patient and within-patient variability using median absolute deviation (MAD) of SIC deviations (methods in Sect G in S1 Text). Granulocyte interactions with TAMs, vasculature, and hybrid E/M cells exhibited the highest heterogeneity, suggesting patient-specific or spatially localized microenvironmental factors (quantitative results in Fig W in S1 Text). Fig 7 shows examples illustrating substantial differences in spatial organization both between patients and across tissue sections from the same patient.

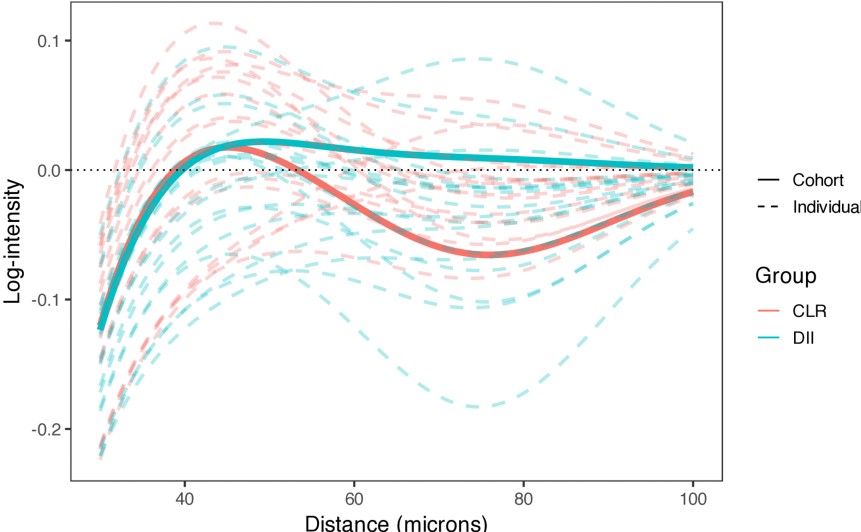

**Fig 6. SIC showing the directional association of CAFs (source) with CTLs (target), stratified by patient group.** Solid lines show cohort-level estimates for CLR and DII; dashed lines show patient-level SICs, illustrating hierarchical variability across patients within each cohort. CTLs in CLR patients exhibit avoidance of CAFs, while DII patients show neutral to clustering patterns, indicating group-specific spatial organization.

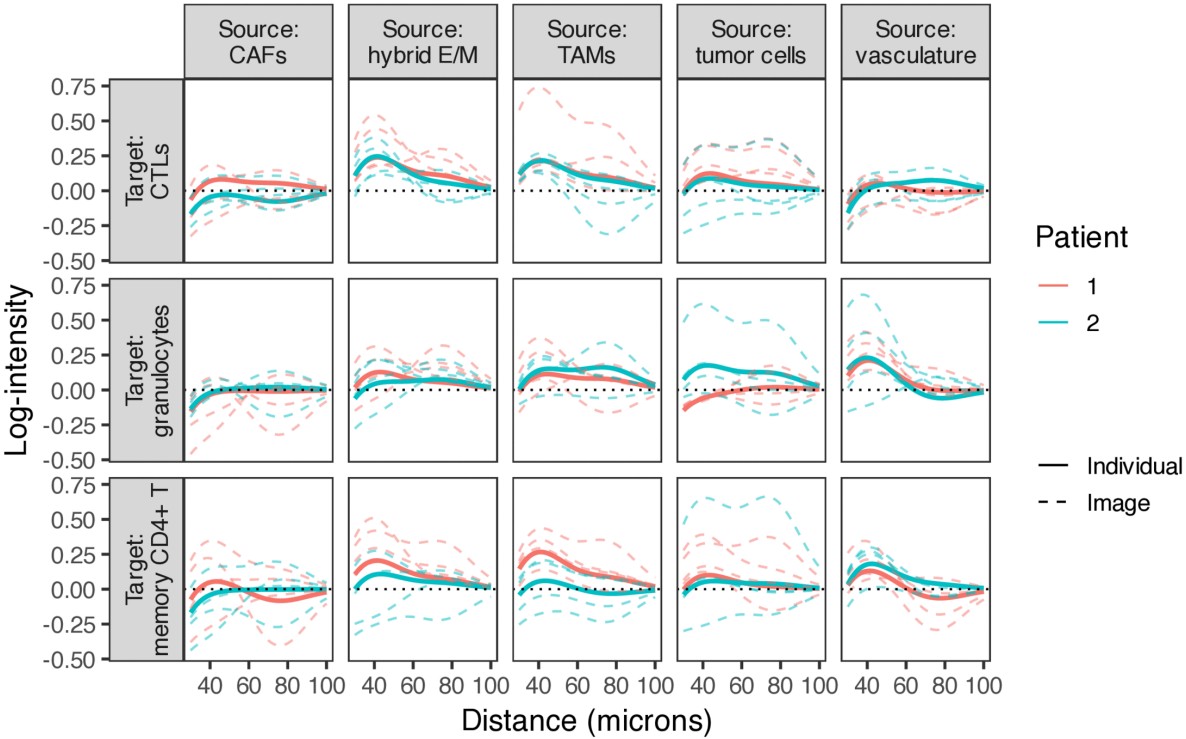

**Fig 7. Examples of patient- and image-level SICs for high-heterogeneity cell type pairs.** Solid lines: patient-level SICs; dotted lines: image-level SICs, illustrating variability within patients.

## 4.4 Comparison of spatial interactions between immune-infiltrated and immune-excluded tumors

Distinct tumor immune phenotypes, such as immune-infiltrated (CLR) versus immune-excluded (DII) tumors, are associated with differential immune activity and prognosis. To investigate whether these functional differences are accompanied by changes in spatial organization, we used SHADE to compare cohort-level SICs across patient groups.

Fig 8 shows the estimated cohort-level SICs for all source-target cell type pairs, stratified by patient group. Visual comparison of CLR and DII curves reveals no statistically significant differences: simultaneous 95% credible bands overlap at all distances for all cell type pairs. While several suggestive trends are visible (e.g., greater CTL-tumor clustering in DII, stronger immune-stromal segregation in CLR), these patterns do not reach significance thresholds and should be considered strictly hypothesis-generating. Detailed exploratory observations and potential biological interpretations are provided in Sect E.3 in S1 Text for readers interested in formulating hypotheses for future validation studies.

To validate that these null differences are not artifacts of tissue architecture confounding—a potential issue identified in our compartment robustness simulations (Robustness to spatial confounding)—we fit compartment-adjusted models incorporating binary compartment covariates based on local tumor cell density. Spatial interaction patterns showed no significant change after compartment adjustment (overlapping credible intervals; Fig Q in S1 Text), confirming that the observed CLR vs DII similarities reflect true spatial organization rather than architectural confounding. Granulocytes showed significant depletion in tumor-enriched regions (particularly in CLR patients), while CTLs and memory CD4+ T cells showed minimal compartment effects (Sect E.2 in S1 Text).

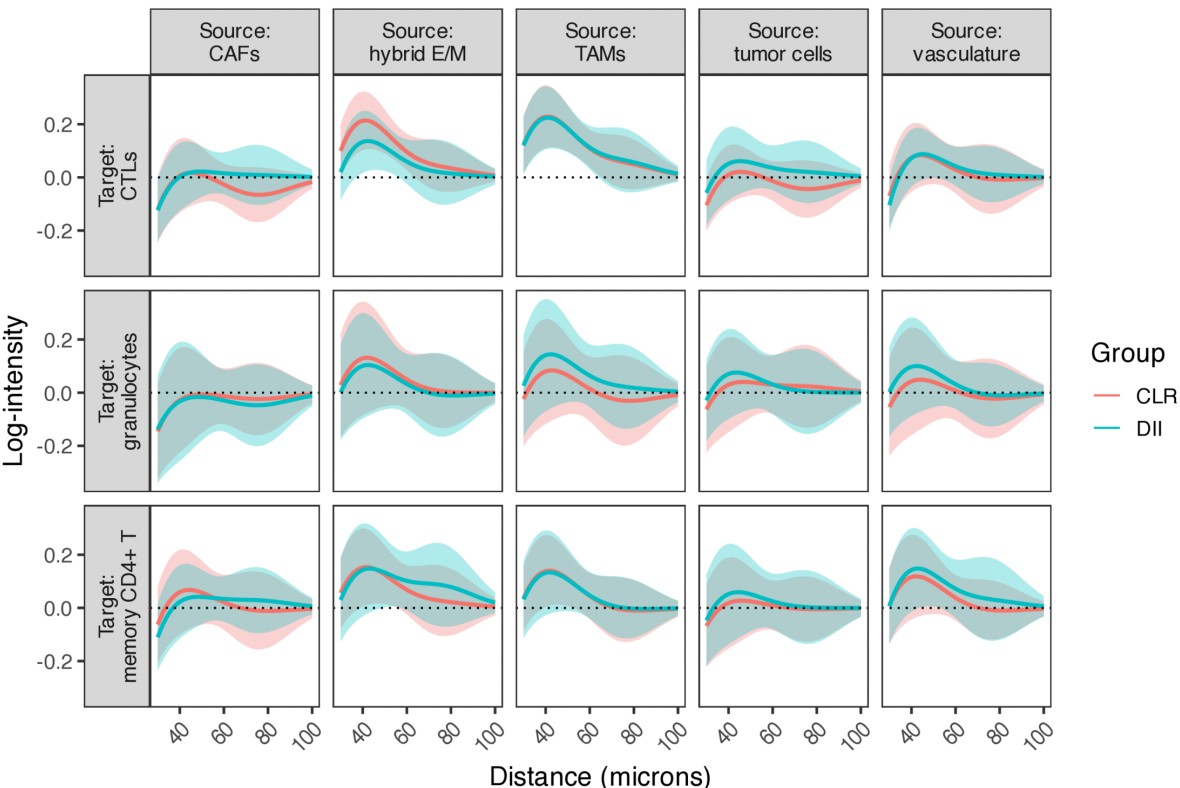

**Fig 8**. Cohort-level SICs ($\psi_{t_1 \to t_2}^{(g,p)}$) estimated for all source–target cell type pairs in the CRC dataset, stratified by CLR and DII patient groups, with simultaneous 95% credible bands.

Beyond estimating spatial interactions, SHADE's conditional intensity functions enable prediction of target cell spatial distributions from source cell configurations. Predictive performance varied by cell type and tumor subtype, with all target cell types showing higher prediction accuracy in CLR versus DII tumors (Sect E.5 in S1 Text).

## 5 Discussion

We introduced SHADE, a Bayesian hierarchical model for quantifying asymmetric spatial interactions in multiplexed imaging data. Through simulation studies, we demonstrated that hierarchical modeling of conditional intensity functions improves estimation accuracy and detection power, particularly under low cell densities and limited sampling, consistently outperforming standard envelope-based methods and non-hierarchical alternatives. A key advantage of SHADE over traditional spatial summary statistics is its ability to formally model heterogeneity across biological scales through hierarchical partial pooling, enabling variance decomposition and more nuanced biological inference than post-hoc image-by-image comparisons.

Application to colorectal cancer revealed substantial spatial heterogeneity across patients and tissue sections, with no significant cohort-level differences between immune-infiltrated and immune-excluded tumors. This highlights the importance of quantifying variability rather than assuming homogeneous spatial patterns. Comparisons with traditional marginal methods (G-cross, functional PCA; Sect E.4 in S1 Text and Sect E.7.3 in S1 Text) revealed concordance for some interactions (e.g., granulocyte-TAM clustering) but divergence for others, where SHADE's multivariate adjustment exposed conditional dependencies obscured in marginal analyses. These approaches are complementary: SHADE provides conditional inference with hierarchical pooling, while functional methods offer model-free characterization of marginal patterns.

A key strength of SHADE is its flexibility in modeling directional interactions across spatial scales, avoiding the restrictive symmetry assumptions of traditional spatial models. Moreover, the use of logistic regression for conditional intensity estimation allows for scalable and stable inference, sidestepping common numerical pitfalls of direct Poisson modeling.

Nonetheless, SHADE has limitations. First, while SICs capture directional spatial association, they remain correlational and cannot determine causality or infer mechanisms of interaction. Second, the SICs reflect spatial dependence rather than molecular signaling pathways, which may limit biological interpretability in some settings. Third, although SHADE supports biologically motivated conditioning structures, exploratory analyses may benefit from modeling both directions of association when directionality is uncertain. Fourth, hierarchical shrinkage produces credible intervals that are narrower than marginal image-level intervals, reflecting the precision-accuracy tradeoff inherent in borrowing strength across images (see Sect D.9 in S1 Text).

Future extensions could incorporate functional covariates (marker intensity, proliferation, exhaustion scores) into the SIC framework, enabling joint analysis of spatial structure and functional state, or leverage cell boundary information rather than centroids to better capture contact-based interactions.

SHADE addresses critical gaps in spatial analysis of multiplexed imaging by enabling asymmetric, hierarchical modeling of cell-cell interactions. The framework provides previously unavailable capabilities: modeling directional relationships, quantifying patient- and image-level heterogeneity explicitly, and incorporating covariates to disentangle spatial associations from architectural confounders. By quantifying directional, multiscale tissue organization, SHADE provides a critical bridge between spatial imaging data and mechanistic understanding of disease biology.

## Supporting information

**S1 Text. Contains supplementary methods (background on Gibbs point process models, logistic regression approximation, simultaneous credible band construction, extended simulation methods), supplementary results (hyperparameter studies, coverage and Type I error analysis, robustness to spatial confounding), extended**

colorectal cancer analysis results, comparison with multilevel functional PCA methodology, and supplementary figures and tables.
(PDF)

## Author contributions

**Conceptualization:** Joel Eliason.

**Data curation:** Joel Eliason.

**Formal analysis:** Joel Eliason.

**Funding acquisition:** Arvind Rao.

**Investigation:** Joel Eliason.

**Methodology:** Joel Eliason, Michele Peruzzi.

**Project administration:** Arvind Rao.

**Resources:** Arvind Rao.

**Software:** Joel Eliason.

**Supervision:** Michele Peruzzi, Arvind Rao.

**Validation:** Joel Eliason, Michele Peruzzi.

**Visualization:** Joel Eliason.

**Writing – original draft:** Joel Eliason.

**Writing – review & editing:** Joel Eliason, Michele Peruzzi, Arvind Rao.

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
