## [Decision Letter · Decision Letter 0]

29 Sep 2025

PCOMPBIOL-D-25-01537

SHADE: A Multilevel Bayesian Framework for Modeling Directional Spatial Interactions in Tissue Microenvironments

PLOS Computational Biology

Dear Dr. Eliason,

Thank you for submitting your manuscript to PLOS Computational Biology. After careful consideration, we feel that it has merit but does not fully meet PLOS Computational Biology's publication criteria as it currently stands. Therefore, we invite you to submit a revised version of the manuscript that addresses the points raised during the review process.

Please submit your revised manuscript within 60 days Nov 29 2025 11:59PM. If you will need more time than this to complete your revisions, please reply to this message or contact the journal office at ploscompbiol@plos.org. Please include the following items when submitting your revised manuscript:

We look forward to receiving your revised manuscript.

Kind regards,

Johannes Textor

Academic Editor

PLOS Computational Biology

Marc Birtwistle

Section Editor

PLOS Computational Biology

**Additional Editor Comments :**

The reviewers were all generally favorable about your work. All reviews are have however raised several points for clarification or improvement that would need to be addressed. If you decide to revise and resubmit your manuscript, please include a point-by-point response to these issues and explain how you addressed them in your revision, or why you chose not to address them.

**Journal Requirements:**

5) Thank you for stating "A.R. serves as a member for Voxel Analytics LLC and consults for Telperian, Tempus Inc. and TCS Ltd." Please modify your 'Competing Interests' statement, and declare all competing interests beginning with the statement "I have read the journal's policy and the authors of this manuscript have the following competing interests:"

**Reviewers' comments:**

Reviewer's Responses to Questions

Reviewer #1: The authors present a Bayesian probabilistic model of the spatial relationship between cell types identified by multiplexed immunofluorescence, and presumably other assays such as spatial transcriptomics. The modeling choices appear sensible, with the spatial distribution of cells of a particular type described by an inhomogeneous Poisson point process whose intensity values are described as a log-linear function of the density of neighboring cells of other types. This method is limited to modeling discrete cell types or states, though it's a flexible enough model that one could imagine extending this to model dependence on marker intensities, which the authors suggest as a future extension. The use of a hierarchical model is an especially nice feature, as analysis of multiple replicates seems to be an afterthought too often in spatial experiments.

With that said, there are a few significant areas I hope the authors will give more consideration to in a future revision:

For inference, the method avoids computing the integral term of the Poisson point process likelihood and substitutes an approximate scheme based on logistic regression classification of observed cells versus homogenous Poisson noise. This is motivated at least in part by computational efficiency concerns, yet there doesn't appear to be any discussion of computational efficiency. Spatial technologies are scaling now to hundreds of thousands of cells per sample, so a natural question the authors ought to address is how this method scales. For example, Equation 3 scales quadratically with the number of cells if implemented naively, which would quickly become problematic for many thousands of cells.

What also frustrates interpretation is that every pair of cells has a negative association at very short distances due to spatial crowding (since cells are not really points, there's a limit to how close their centroids can be). Since most of the spatial associations detected appear to be pretty close range ones, it makes negative associations in particular hard to determine. It's not obvious at what distance we should interpret negative numbers as due to crowding versus due to cellular organization. It wasn't clear to me why there couldn't be a term in the regression to capture this generic crowding effect. Maybe more ambitiously, if full segmentation boundaries are available, cellular distances could be computed as boundary-to-boundary avoiding this issue.

Some of the positive interaction scores (e.g in Fig. 6 and 8) seem quite small. Since inference is done via sampling, their credible intervals should be available which might tell us what we should consider significant. Fig. 3 does appear to show some type of uncertainty region, though I don't see a description of what exactly it is. More broadly, I'd like to see some discussion of assessing the overall statistical significance of associations. I would imagine a common use case of SHADE is to test all pairs of annotated cell types in both directions. Should one then inspect plots of all n^2 pairs in both directions, which could be hundreds of pairs, or is there some way to compute an aggregate posterior probability of non-zero effects at some distance?

Minor issues:

Some acronyms are used without being defined (e.g. CTLs, TAMs). These are common immunological terms but still should be defined to avoid ambiguity.

Figure 10 caption needs more explanation. I'm guessing the black dots are observed cells of the labeled type, but it's not clear.

Reviewer #2: The manuscript by Eliason et al. describes a method and software tool for quantitative analysis of point patterns describing cell types derived from multiplexed imaging data. The main output of the tool is the "spatial interaction curve", a distance-dependent function of the (directional) intensity of the influence of one cell-type on another regarding spatial positioning, e.g. co-localization of immune cells with tumor cells. The method is demonstrated using both synthetic data and application to a suitable published data set.

The described method appears to be a valuable contribution to a research field that indeed is short on methods with quantitative output suitable for systematic evaluation of large data sets (e.g. patient cohorts). The paper is well-written and the applications are well-chosen. Nevertheless, a number of points are not quite clear to me, especially regarding interpretation and statistical analysis of SIC curves, and I am a bit concerned on whether the analysis of synthetic data really adresses the most relevant points - please see details below with reference to text fragments:

- Introduction: "... yet existing spatial models typically analyze images independently, potentially overlooking heterogeneity across patients", and similarly abstract: "Many analysis methods ... treat images independently".

Generally, I feel such statements should be made explicit and substantiated with references. Also, I do not quite understand the point, why would analyzing images independently automatically mean that patient heterogeneity is not studied? And is Shade not actually based on computing statistics (SIC curves) on individual images? In the current form of the ms, I do not find any follow-up on this point in the results or discussion sections, please clarify.

- Methods (p. 8), "The SIC summarizes ... This curve represents the expected contribution of type A_k cells to the log-intensity...".

The term "log-intensity" appears for the first time at this point, and lacks a direct interpretation. What does log-intensity of a cell type physically mean, is it related to the spatial cell density? Or would it be possible to normalize it to the average cell-density in the image, or a set of images? This is critical for interpretation of the SIC curves where log-intensity appears on the y-axis throughout the paper.

- Methods (p. 9), "A strong A->B SIC indicates that the presence of A is statistically predictive ..." (and similar statements in the following).

It is not clear to me what statistically predictive means here. Is it possible to construct a statistical test for SIC curves different from spatial randomness, or for SIC curves differing between different image sets? The next section about "Multilevel Bayesian Model" suggests that the authors aim for statistical analysis, but I have trouble finding any clear statistical statements, also in the following chapters where synthetic data and image data sets are analyzed. Figures 4 and 5 show confidence intervals but statements on statistical significance are missing, I suggest evaluating significance using bootstrapping procedures. In Figure 4, it seems that all sown curves are within the confidence of the null (CSR) model, does that mean that the generated synthetic data cannot be distinguished from CSR?

- In the same context, it would make sense to include more methods for comparison (in addition to G-cross), for example Ripley's K-function is mentioned in the introduction.

- Simulation Studies (p. 12), "Spatial patterns were simulated ... with source points generated from a homogeneous Poisson process and target points influenced by spatial interaction curves ..."

This sounds to me as if the synthetic data were generated under exactly the same assumptions used for deriving the SIC curves proposed for analysis, is that correct? Would that not by construction give an advantage to the method proposed by the authors when comparing different analysis methods? It would be important to test the method also using other sources of synthetic data, e.g. starting from Bridson sampling or using a Gauss filter after the Poisson process.

Minor points:

- Introduction, "While our approach is not a direct extension of the multitype Gibbs points model ..., it is certainly inspired by it" (and similar statements in Methods): It would be very helpful to formally introduce these methods the paper is based on, possibly in a supplement, and then show direct comparisons on a suitable test problem. At the moment, it is not straight-forward to evaluate the extensions and modifications introduced by the authors.

- The introduction is quite long and contains sections that might be better placed in the discussion, such as specific comparison to other methods.

- Methods (p. 7), "... assuming that ... follows an inhomogeneous Poisson point process": Could the authors indicate what that assumption implies in the context of typical applications of the new method (e.g., a Poisson process typically is based on a rare-events argument)? And would there be reasonable alternative choices for the probability distribution?

Reviewer #3: This manuscript presents a method for can detect spatial interactions (co-localizations) that are directional (asymmetrical). The method allows for modeling jointly multiple images simultaneously from multiple subjects and cohorts within a Bayesian hierarchical model. It also allows for the assessment of these interactions at a range a distances using a curve (spatial interaction curve, SIC). The key innovation of this work is the asymmetric aspect and the ability to measure interactions as various distances and not one pre-specified distance. They compare the method to G cross and apply to a large colon cancer study published in Cell in 2020. The manuscript was well written. Below are my comments/questions about the approach and presentation of results.

1. How does the SIC compare to using functional data analysis (FDA) of the spatial curves that ones gets from K or G? For example, using basis functions that look similar in formulation to functional data analysis and functional principal component analysis?

2. It would be good to compare you approach to using FDA on the G or K statistics, as implemented in the R package mxfda(Wrobel et al. 2024). Additionally, should compare to not just G but also K as this has been used often in practice and has been reported to have better discrimination ability for detecting co-localization than G(Soupir et al. 2025).

3. In the set-up of the model and approach in section 2.1, assume a inhomogeneous Poisson process. However, in the computational approximation using logistic regression you have a homogeneous Poisson process. In the logistic regression model fitting, how do you account for inhomogeneity? That is, is the analysis done within the tumor and stroma compartments of the tissue as different cells could have different intensity within these two tissue domains. If this is not explicitly accounted for in the model, I would recommend that analysis be done by tissue domains to limit the issue of inhomogeneity.

4. In the simulation study, please state how many simulated datasets were generated for each scenario. I think you have 30 simulations per scenario but not sure. If this is the case that used only 30 simulations per scenario, why so few of simulations? Possible to have 100 per scenario to get more precise estimates of power and type I error rate to compare the 3 approaches? Also, state in the simulation set-up the range of cell abundances that assessed and why 39 simulated used to estimate envelopes for G.

5. Please have a “null” scenario and present the type I error rate. In Figure 5 only presenting the power which can’t be interpreted without knowing the approaches control the type I error rate.

6. Please present computational time to fit the model. The colon cancer data (I believe) is based on a TMA. How would this method scale for computing on whole tissue slides with millions of cells?

7. In the hierarchical model (section 2.2), please present where in the model you assess for differences by a factor/exposure (i.e., CLR and DII as used in the colon study).

8. Please provide details on the analysis of the colon cancer data how you determined if the clustering was significant taking into account random chance? I don’t see any information presented on how this would be done beyond just reference to figure that shows slightly different curves (Figure 6, Figure 9). In figure 9, looks like set arbitrary level at a score of 0.05.

9. G assumes homogeneity in the point process. Please look at results for G by tissue compartment (tumor, stroma, etc) to somewhat control for difference in intensity of different cell types by tissue domain.

10. What does the MAD presented in Figure 7 look like between CLR and DII tumors?

11. A better literature review related to methods for co-localizations and those applied to mIF data in cancer should be in the introduction of the paper.

Minor:

1. Figure 5: I can’t seem to see the 20, 40, and 60 nm data. Also, add to caption what is hight and low levels.

Soupir, A. C., I. V. Gadiyar, B. R. Helm, C. R. Harris, S. N. Vandekar, L. C. Peres, R. J. Coffey, J. Wrobel, S. Ma, and B. L. Fridley. 2025. 'Benchmarking Spatial Co-Localization Methods for Single-Cell Multiplex Imaging Data with Applications to High-Grade Serous Ovarian and Triple Negative Breast Cancer', Stat Data Sci Imaging, 2.

Wrobel, J., A. C. Soupir, M. T. Hayes, L. C. Peres, T. Vu, A. Leroux, and B. L. Fridley. 2024. 'mxfda: a comprehensive toolkit for functional data analysis of single-cell spatial data', Bioinform Adv, 4: vbae155.

**Have the authors made all data and (if applicable) computational code underlying the findings in their manuscript fully available?**

Reviewer #1: Yes

Reviewer #2: Yes

Reviewer #3: Yes

PLOS authors have the option to publish the peer review history of their article (what does this mean?). If published, this will include your full peer review and any attached files.

Reviewer #1: **Yes:** Daniel C. Jones

Reviewer #2: No

Reviewer #3: No

**Figure resubmission:**
---

## [Decision Letter · Decision Letter 1]

19 Jan 2026

Dear Dr. Eliason,

We are pleased to inform you that your manuscript 'SHADE: A Multilevel Bayesian Framework for Modeling Directional Spatial Interactions in Tissue Microenvironments' has been provisionally accepted for publication in PLOS Computational Biology.

Best regards,

Johannes Textor

Academic Editor

PLOS Computational Biology

Marc Birtwistle

Section Editor

PLOS Computational Biology

Thank you for carefully addressing all reviewer points. Two of the reviewers now recommended acceptance of the manuscript; while the third reviewer was unable to assess your manuscript again, you have thoroughly revised the manuscript in response to their comments as well, and explained the changes clearly in your point-by-point response. Therefore, I would like to recommend acceptance of your manuscript at this stage.

Reviewer's Responses to Questions

**Comments to the Authors:**

Reviewer #1: I continue to think this is an interesting, novel method, that fills an unmet need: multilevel modeling of direction spatial cell-type effects that can account for batches, samples, and conditions.

In my original review, I raised some questions about whether the inference remains tractable for datasets at the scale that are now being generated. The inclusion of timing data in this revision assuages this concern, as the inference seems to be very efficient.

My other two major concerns regarded interpretation of the data. In this revision, the authors address these concerns proposing some convenient tools protocols for exploratory analysis. I remain somewhat concerned about the difficulty of separating the biological effects from spatial effects due to physical crowding. The authors address this with a cutoff, which I think is a fine ad hoc solution, but hopefully after publication the method can further developed with a more principled way to separate out these effects in the point process.

Reviewer #2: The authors have adressed my comments, and I think the paper improved a lot in revision! In addition to the proposed method as such, it is a nice resource for advanced methods on point-pattern analysis incl. synthetic data generation and stastical comparison. I could not check the source-code in detail but it seems to be well-organized in the listed repos.

**Have the authors made all data and (if applicable) computational code underlying the findings in their manuscript fully available?**

Reviewer #1: Yes

Reviewer #2: Yes

PLOS authors have the option to publish the peer review history of their article (what does this mean?). If published, this will include your full peer review and any attached files.

Reviewer #1: **Yes:** Daniel Jones

Reviewer #2: **Yes:** Kevin Thurley

---

## [Editor Report · Acceptance letter]

PCOMPBIOL-D-25-01537R1

SHADE: A Multilevel Bayesian Framework for Modeling Directional Spatial Interactions in Tissue Microenvironments

Dear Dr Eliason,

I am pleased to inform you that your manuscript has been formally accepted for publication in PLOS Computational Biology. Your manuscript is now with our production department and you will be notified of the publication date in due course.

With kind regards,

Anita Estes
